# Improved Proactive Routing Protocol Considering Node Density Using Game Theory in Dense Networks

**Omuwa Oyakhire ***  **and Koichi Gyoda**

Graduate School of Engineering and Science, Shibaura Institute of Technology, 3-7-5 Toyosu, Koto-ku, Tokyo 135-8548, Japan; gyoda@shibaura-it.ac.jp
***  Correspondence: nb18103@shibaura-it.ac.jp

**Abstract:** In mobile ad hoc networks, network nodes cooperate by packet forwarding from the source to the destination. As the networks become denser, more control packets are forwarded, thus consuming more bandwidth and may cause packet loss. Recently, game theory has been applied to address several problems in mobile ad hoc networks like energy efficiency. In this paper, we apply game theory to reduce the control packets in dense networks. We choose a proactive routing protocol, Optimized Link State Routing (OLSR) protocol. We consider two strategies in this method: *willingness_always* and *willingness_never* to reduce the multipoint relay (MPR) ratio in dense networks. Thus, nodes with less influence on other nodes are excluded from nomination as MPRs. Simulations were used to confirm the efficiency of using our improved method. The results show that the MPR ratio was significantly reduced, and packet delivery ratio was increased compared to the conventional protocol.

**Keywords:** multihop networks; OLSR; game theory; mobile ad-hoc networks; willingness; multipoint relay (MPR)

## 1. Introduction

Mobile ad-hoc networks (MANETs) are autonomous networks as they are self-organizing and self-configuring networks where nodes can move independently and relay traffic to other nodes [1]. MANETs have laid a great foundation for sensor networks and internet of things. New applications of these networks are emerging rapidly. It is projected that connected internet of things (IoT) devices would increase to 75.44 billion worldwide in 2025 [2]. Thus, there will be a lot of connected wireless devices per unit area resulting in a dense network. A dense network is a network with numerous nodes in a small area at the same time, and a node belongs to the dense network if its node density (that is, the average number of its one-hop neighbours) is higher than a threshold and almost invariant during relatively long time intervals [3]. A dense network implies a high control overhead as all the nodes will send control traffic intermittently, thus may deplete the battery power of the node or result in a packet loss when application traffic is sent.

Recently, game theory has provided a useful solution in communication networks [4]. It has also modeled and addressed several problems in MANETs, especially in energy efficiency and selfish nodes [5,6]. In game-theory modeled networks, the players (nodes) have conflicting objectives; thus, a utility function representing a reward that allows each player to evaluate an outcome that reflects on its objectives. The utility is based on the player's strategies but also on other players' actions.

In this same context, our research is focused on minimizing the multipoint relay (MPR) ratio and redundancy in dense networks. We propose to change the MPR willingness of a node based on its node

density. All nodes will participate in a game, and the node density of each node (number of one-hop neighbours) will determine its *node_willingness*. In the heuristic Optimized Link State Routing (OLSR), the node willingness has a value of WILL_DEFAULT. By changing the node willingness of each node due to the influence of its neighbouring nodes, we can improve the MPR selection, particularly in dense networks. This improves the overall performance of the routing algorithm. Therefore, our contribution is briefly summarized as follows:

i.　　　First, we propose an algorithm based on game theory. It will determine the willingness of a node to be MPR. In this proposed algorithm, nodes with more influence on other nodes would have its willingness value increased.

ii.　　All HELLO messages are indexed to avoid duplicate messages during node count.

iii.　Thirdly, the key novelty of this paper is that the number of one-hop neighbours of a specific node will determine the node willingness.

The remainder of the paper is organized as follows: In Section 2, it summarizes related studies and indicates where the present research fits in relation to these studies. Section 3 describes our contribution, improved MPR selection for OLSR using game theory. In Section 4, we present the simulation results that show the efficiency of our proposed algorithm. Finally, Section 5 gives the conclusion.

## 2. Related Work

### 2.1. Improvement of OLSR in Dense Networks

The OLSR has the advantage of optimizing the existent bandwidth usage and is suited for high-density networks. OLSR routing protocol also has some limitation. It is proved that the calculation of the minimum MPRs is a nondeterministic polynomial time (NP)-complete problem [7–9]. Thus, there are several extensions to improve the MPR selection. However, for this paper, we focused on the minimization of the MPR sets in dense networks. Table 1 summarizes the related works.

Kitasuka et al. [10] proposed using shared MPR sets to achieve a more efficient MPR selection in moderately dense multihop networks than the conventional MPR selection. They analyzed moderately dense networks to show that a node close to the two-hop border had a small probability of being a two-hop neighbor. Further, they noted there is a chance of the node's MPRs being shared with its neighbors, thus the proposal of shared MPR sets. However, the authors did not investigate node mobility, which is crucial in wireless ad-hoc networks as they simulated only a static environment due to the high computational complexity of their algorithm.

The battery level of the node is another consideration for the MPR selection, as proposed by De Rango et al. [11]. They proposed a novel energy aware MPR election policy, denoted by Energy Efficient OLSR, that allows the energy node to be preserved for a longer time in a dense network. In the Energy EfficentOLSR, each node calculates its own energetic status and then declare an appropriate willingness. The willingness of the node is based on the battery capacity and the predicted lifetime. Similarly, Suhaimi et al. [12] considered the implication of a wireless node's power status and its MPR willingness. They changed the willingness of a wireless node based on its power status. This paper, however, used a limited number of wireless devices thus could not make adequate comparison with existing networks.

**Table 1.** Comparative study of related work.

| References | Objective of the Work | Advantages | Disadvantages | Metric Studied |
|---|---|---|---|---|
| Kitasuka et al [10] | MPR shared sets to reduce routing overhead in dense networks. | Significantly reduced the MPR ratio and MPR and topology control redundancy. | Only considered a static environment. No mobility model was used. | MPR ratio, MPR and TC message redundancy, number of (MPR's, TC messages, OLSR packets, total packets) per node. |
| De Rango et al. [11] | Preserve a node's energy in a dense network. | Network life time increased. | Residual battery power for route computation not considered. | Network throughput, expiration time, average node energy, node lifetime. |
| Suhaimi et al [12] | Analyze the relationship between the battery power and the willingness of a node in OLSR. | Changed the willingness based on the node's battery life. Performed a test-bed experiment. | Limited number of devices. Did not consider mobility. | Willingness, battery life. |
| Yamada et al. [13] | Reduce MPR redundancy by using a cooperative MPR selection. | Topology control packets were significantly reduced. | Node density was not considered. | Topology Control packets, arrival time of pinged packets. |
| Kumar et al. [14] | Improve OLSR in networks with high mobility. | Significantly reduced control overhead and more link bandwidth for a dynamic topology. | Considered a minimal network load. | Packet Delivery ratio, routing overhead, End to end delay, throughput. |
| Sanguankotchakorn et al. [15] | Reduce control overhead by nodes deciding to update or not update its HELLO and TC messages. | Reduced control overhead while average throughput remains unchanged. | Average throughput was decreased. Packet loss was not considered. | Routing overhead ratio, throughput, normalized overhead ratio. |

Yamada et al. [13] proposed a cooperative MPR selection procedure to reduce topology control (TC) packets in dense networks. They showed that they are redundant control messages which can be piggybacked with other control messages in dense networks. They aimed to find a minimal set of generators and forwarders of TC messages, without modification of the properties of multipoint relays. Using their cooperative MPR selection procedure, some nodes selected their MPR independently, by the conventional MPR selection while other nodes referred to the MPR selection of neighbors, to make TC message senders small. Thus, the network reachability is kept similar to that of the conventional MPR selection procedure.

Kumar et al. [14] proposed considering the position of the source and destination nodes while selecting MPRs. They showed that the control overhead was significantly decreased with their new routing scheme, Airborne-OLSR (AOLSR) for use in airborne ad-hoc networks. Due to the high mobility of these devices which result in frequent topology updates, the MPR selection is chosen on either the left or the right side of the source node to which the data is to be sent. Despite, this is used for high mobility networks like Flying Ad-hoc Networks (FANETs), it could also be applicable for sensors and IoT networks.

Sanguankotchakorn et al. [15] proposed using game theory to reduce routing overhead in OLSR. They proposed all nodes play a game to decide to update or not update its HELLO and TC messages in order to minimize performance degradation and waste of resources. After an elapsed time, all nodes chose a strategy dependent on the node's speed and movement probability. The normalized routing overhead ratio was reduced but packet loss was not considered.

### 2.2. Position of This Research

As mentioned in Section 2.1, existing studies mainly improved the MPR selection or the node willingness in OLSR networks. They extended the MPR selection algorithm in dense networks, changed the node willingness to preserve its energy in dense networks, change the node willingness based on the battery life and reduce the MPR redundancy using shared sets. However, changing the node willingness based on the HELLO messages received has not been proposed. The position of this present research relative to existing studies is considering mobile dense networks and minimizing the MPR ratio.

## 3. Materials and Methods

We describe the materials and methods for improving the Optimised Link State Routing protocol based on game theory.

### 3.1. Node Willingness

The core functionality of OLSR [7,8] is the mechanism of optimizing topology control traffic based on the multipoint relay algorithm, as proposed by Qayyum et al. [16]. If a node is selected as an MPR node, there is always a path to its two-hop neighbours. The choice of MPR in OLSR affects routing performance. It is proved that the calculation of the minimum MPRs is an NP-complete problem thus to calculate the MPR set is very difficult, and some heuristic algorithm must be used to find the approximate optimal solution [17]. OLSR uses HELLO messages to find its one-hop neighbours and its two-hop information. Also, it performs a distributed election of a set of multipoint relays (MPRs). In the HELLO message packet format as seen in Figure 1, there is a willingness field. It specifies the willingness of a node to forward traffic for other nodes which is a criterion for the MPR selection. The willingness of a node can assume any integer value from 0 to 7. Nodes, by default, have a willingness of WILL_DEFAULT. A node can change its willingness value as its conditons change, for example, being low on battery or permanent power or high capacity interference to other nodes. Table 2 shows the degree of willingness and its corresponding values.

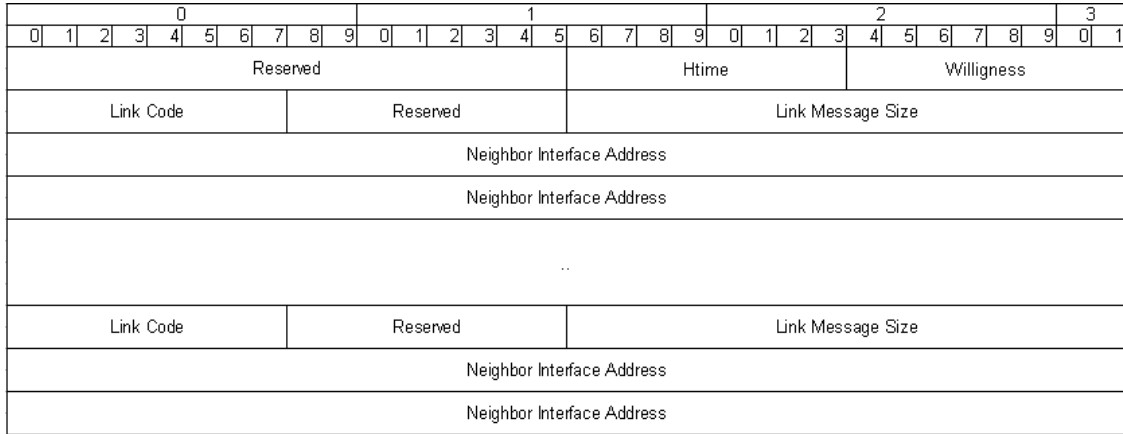

**Figure 1.** HELLO message packet format in an OLSR network [8].

**Table 2.** Willingness and its values [8].

| Degree of Willingness | Value |
| --- | --- |
| WILL_NEVER | 0 |
| WILL_LOW | 1 |
| WILL_DEFAULT | 3 |
| WILL_HIGH | 6 |
| WILL_ALWAYS | 7 |

### 3.2. OLSR Improvement Selection

In this section, we propose an algorithm to change the node willingness based on the HELLO messages received after an elapsed period using a game approach before the MPR selection.

#### 3.2.1. Game Mathematical Model

In this section, we propose describing the ad-hoc network using the game theory approach [18]. It is represented as:

$$G = [N, \ S, \ U] \tag{1}$$

where N is the set of nodes in the network, S is the strategy set and U is the utility function. Table 3 shows the similarity between a game and a wireless ad-hoc network.

**Table 3.** The duality of game and wireless ad-hoc network.

| Elements of a Game | Elements of the Wireless Ad-Hoc Network |
| --- | --- |
| Players | Node in the network |
| Strategy | We consider two strategies: <br> i. *willingness_always* <br> ii. *willingness_never* |
| Utility function | The node count after an elapsed period |

In this network, each node will receive the utility function and then evaluate a particular outcome depending on its strategy. All nodes aim to maximize or minimize the utility function. The utility function of the $i$th node is shown in Equation (2).

$$U_i = \left\{ \begin{array}{ll} G - f(c_i); & if \ S_i = willingness\_always \\ 0 & if \ S_i = willingness\_never \end{array} \right\} \tag{2}$$

where G is the gain received by nodes that the terminal count is above a certain threshold and f (c) is the cost function paid by node *i* which changes its willingness value. The cost function is assumed to be constant for all nodes in the network. Then, the utility matrix for any node is illustrated in Table 4.

**Table 4.** Utility function of the *i*th and *j*th node.

| Node *i* | Node *j* | |
| --- | --- | --- |
| | *willingness_always* | *willingness_never* |
| *willingness_always* | G-c | G-c |
| *willingness_never* | G | 0 |

We use the mixed strategy of the Nash equilibrium [19]. If the probability of a player to choose *willingness_always* is *p*, then the probability to choose will_never is 1-*p*. The probability of a player to choose *willingness_always* will be

$$p = 1 - \left(\frac{c}{G}\right)^{\frac{1}{N-1}} \tag{3}$$

The probability is valid for the mixed strategy of the Nash equilibrium. The probability *p* depends on N, c and G. As N and c are constant, then *p* depends on G. The gain is dependent on the speed of the node at time *t* and the movement probability, which is based on the Random Waypoint model. The utility function is dependent on the number of nodes after an elapsed time. The maximum number of one-hop neighbours is set to be the threshold, T. This threshold value is the number of HELLO messages received from a node's one-hop neighbours which we call *node_density*. The threshold is set at a value of 0.25 of the total nodes in the network. This value was achieved after a series of iteration [20]. Each node will compare its value to the threshold and choose its strategy as follows:

if *node_density* $\geq$ T then *willingness_always* or *node_density* $<$ T then *willingness_never*

In this paper, we consider a non-cooperative game where the players maximize the payoff and choose the best strategy individually and rationally.

### 3.2.2. Enhanced Node Willingness Algorithm

In this section, we present the algorithms for the improved OLSR algorithm in multihop networks. Figure 2 shows the flowchart for the node willingness based on the node count of each node. All nodes in the network will go through this additional process when sending HELLO messages. Figure 3 is the flowchart for the indexing of the HELLO messages received. Each node will index the HELLO message received and process the IP address of the node. Then, using this unique index, all the one-hop neighbour nodes are counted which will be used for the node willingness.

Both algorithms (Algorithm 1 and Algorithm 2) modify the MPR selection algorithm. The rest of the OLSR algorithm remains the same.

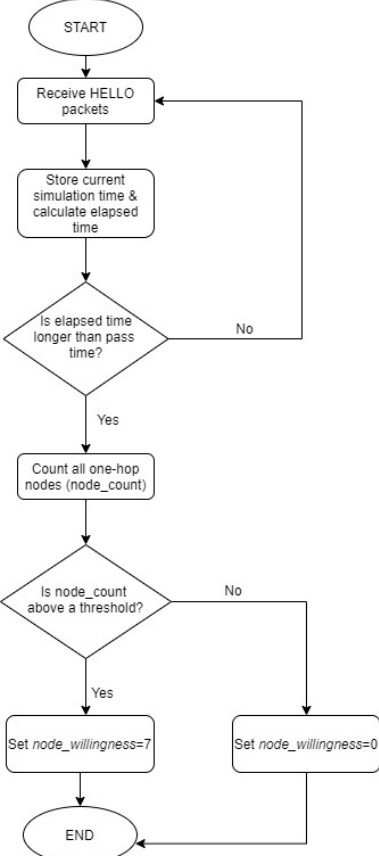

**Figure 2.** Node willingness based on the node count.

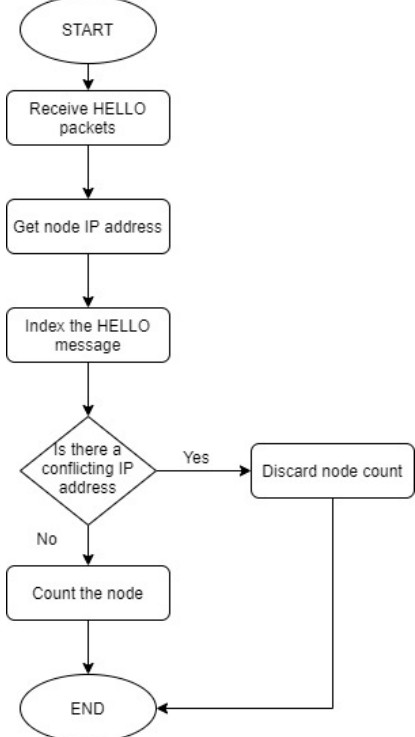

**Figure 3.** Indexing the HELLO messages and counting the corresponding node.

---

**Algorithm 1: Node willingness of nodes to become MPR**

---

**Improved OLSR algorithm considering node willingness and node count.**
Let $t_1$: Lower bound time; $t_2$: Upper bound time; $T$ is the threshold of node density.
sim-time[$x$]: Arrival time of hello packet [$x$]; sim-time [0]: Initial time of hello packet is sent.

1:      **Define** Pass time = sim-time[$x$]-sim_time[0].
2:          **if** (pass time < $t_1$)
3:          **then** increase the sim-time[a]
4:      **else** count the nodes (terminal_count)
5:          **if** ((sim_time[a]> $t_2$) && (terminal_count<= $T$))
6:              **then**:
7:          node_willingness=WILL_NEVER;
8:      **else** node willingness = WILL_ALWAYS.

---

---

**Algorithm 2: Index all Hello messages received by each node**

---

1.    Process all Hello messages received from each node.
2.    Copy the IP address of the node and index the node.
3.    Repeat this for all hello messages received.
4.    If another hello message is received from the same node, compare its IP address and index, if it is the same, do not count the node.

---

### 3.2.3. Proposed Example

Let us consider a highly dense network where the source node wants to send packets to the destination node. According to our algorithm, all nodes will calculate the number of HELLO messages received from its neighbours. This is denoted as the node_density. If the node_density is above a threshold, then the node willingness is set to WILL_ALWAYS. However, if the node_density is below or equal to the threshold, the node willingness is set to WILL_NEVER. For example, as seen in Figure 4, if the threshold is 5, the node on the left willingness is increased to 7 while the one on the right is set to 0. Thus, only nodes that have more one-hop neighbours has its MPR willingness increased to 7 and those with lesser one-hop neighbours has its willingness set to a value of 0. This game is played every 5 seconds in each node. Every node has to update its willingness value based on the terminal count, which is the number of HELLO messages received from its one-hop neighbours

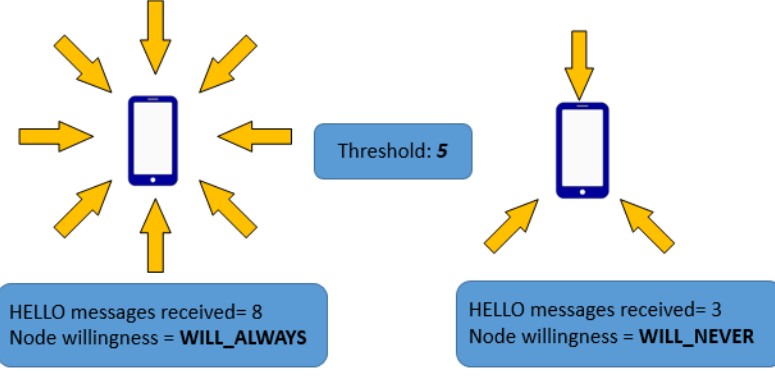

**Figure 4.** Node willingness selection based on the number of received HELLO messages.

## 4. Simulation Environment and Results

In this section, we describe our simulation setup and present the evaluation results.

*4.1. Simulation Parameters*

We evaluated the performance of our improved protocol by doing simulations using the Riverbed Modeler 18.0 [21]. Riverbed Modeler simulates real network environments and analyzes the network. Furthermore, existing protocols can be enhanced and evaluated easily. With a bid to make this network suitable for unstable and dynamic environments, we included random behaviour in all nodes except the source and destination node. Due to the mobility of the nodes, the dynamic receiver group computation is set to refresh every 10 s [22]. The transmission range is set uniformly at 30 m to ensure a longer multihop from the source to the destination.

At the beginning of the simulation, all nodes are randomly deployed in a 100 m × 100 m area and are free to just within this area. We set the traffic packets to simulate a simple voice codec, G.711 at 8 KB/s. Thus, a 6400-bit packet is sent every 0.1seconds from the source node to the destination node. The pause time is set to zero so that the node keeps on moving during the simulation. The simulation time is set to 600 s. The HELLO traffic and TC message intervals are set to 5 seconds and all nodes are set to WILL_DEFAULT willingness and then changed after the game is played. User datagram protocol (UDP) is used as the transport protocol. Every scenario simulates 10 minutes of network activity. The simulation is carried out 10 times using various random positon of the nodes. Table 5 summarizes the simulation settings.

**Table 5.** Simulation parameters.

| Parameter | Value |
| --- | --- |
| Network Simulator | Riverbed Modeler 18.0 |
| Simulation time | 600 seconds |
| Number of nodes | 25, 50, 100, 150 and 200 |
| Protocol used | OLSR, Improved OLSR |
| Nodes deployment | Random |
| Number of seeds | 10 |
| Transport protocol | UDP |
| Transmission range | 30m |
| IP | IPv4 |
| Physical layer method | PHY 802.11n |
| Transmit power | 0.0015W |
| Transmission range | 30m |
| Packet size | 8 kilobyte/sec |
| Node speed (m/s) | 0–1 |
| Pause time | 0 seconds |

*4.2. Evaluation Metrics*

We use four metrics to explain our simulation results. The first two metrics concern the routing protocol. Also, the packet delivery ratio and end to end delay metrics are measured.

- MPR ratio: This is the ratio of MPR's to the number of the nodes in the network.
- Routing overhead ratio: This is the total number of routing packets over the total number of received data packets.
- Packet delivery ratio (PDR): This is the ratio of the data packets successfully delivered to the destination compared to the packets sent from the source
- End-to-end delay: This refers to the time taken by a packet to be transmitted through any network from the source to the destination.

*4.3. Simulation Results*

For each scenario, we ran 10 simulations with different random mobility scenarios. We calculated the average result with a confidence interval of 95%. Figure 5 shows the simulation result for each

metric. In each figure, the results of the conventional OLSR and the improved OLSR are shown with the labels "OLSR" and "Improved OLSR", respectively.

The plot in Figure 5a shows the MPR ratio. Wu et al. [23] have proved shown that this ratio will eventually increase to 1 in dense networks. By comparing the ratios of the conventional OLSR with the improved OLSR, we see that there is over 5% redundancy in the low-density networks of 100 nodes and less. In highly dense networks, the redundancy is slightly reduced, (1% in 150 nodes and 2% in 200 nodes). A comparison with [10] shows that the improved and conventional MPR ratio is higher. However, Kitasuka et.al only considered a stationary scenario compared to our highly dynamic scenario.

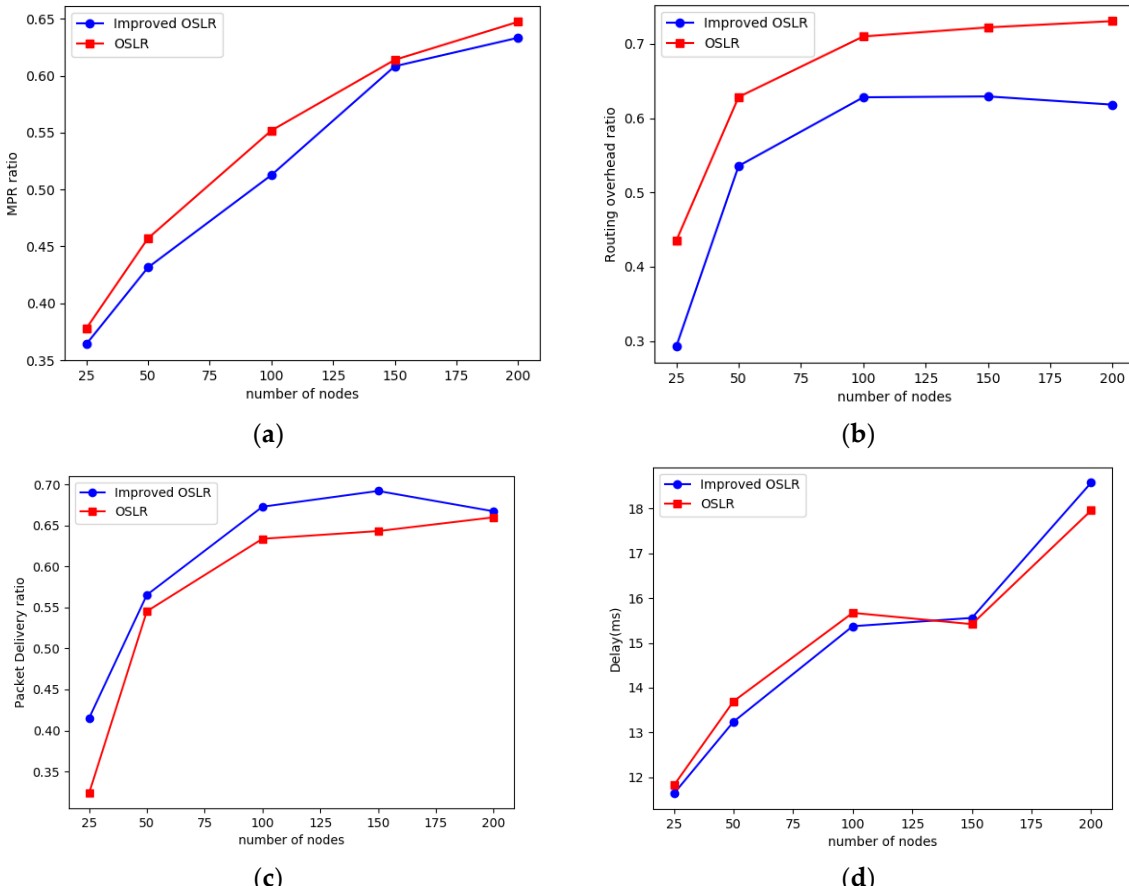

**Figure 5.** These results show the simulation results comparing OLSR and the improved OLSR protocols. (**a**) MPR ratio; (**b**) Routing overhead ratio; (**c**) Packet delivery ratio; (**d**) Average end-to-end delay.

Figure 5b shows the routing overhead ratio as a function of the number of nodes. The routing overhead is the ratio of total routing packets (inclusive of HELLO packets and topology control packets). There is a significant decrease in the routing overhead ratio. This is due to fewer MPRs selected. This implies that there will be enough bandwidth for the application data packets.

Figure 5c shows the relationship between the packet delivery ratio and the number of nodes. The packet delivery ratio is a parameter that indicates the success of the routing protocol. In low-dense and moderately dense networks, there is a significant reduction of the packet delivery ratio. In the 25-node scenario, the PDR is below 45% in both improved and conventional OLSR. The low PDR is due to the high mobility of all nodes in the network that implies that the route link has to be calculated often; however, there are a few alternate links, thus resulting in a high packet loss. In moderately dense and highly dense networks, there are more alternative links despite the mobility, thus a higher PDR. There is a significant increase in the packet delivery ratio except in 200 nodes where the increase is 1%. OLSR protocol has a minimal delay due to the routes being available when needed.

Figure 5d shows the relationship between delay and the number of nodes. OLSR as a proactive protocol has a minimal delay, as the routes are available when required. As shown in Figure 5d, the average end-to-end delay is less than 20 ms in all scenarios that will be barely noticeable. In all scenarios, there is no significant difference in improved OLSR and OLSR. Thus, we can conclude that the delay is negligible.

Here, a comparison of our improved protocol is compared with Kitasuka et al. [10]. In both cases, the MPR ratio is minimized in dense networks. Kitasuka et al. propose shared MPR sets to minimize the MPR ratio. Our proposal uses the node density to reduce the MPR ratio. We have compared these two methods and found that both methods significantly reduce the MPR ratio. It is found that our method has a slight difference in highly dense networks.

## 5. Conclusions

In this paper, we proposed an algorithm to improve the Optimized Link State Routing protocol in mobile scenarios. This algorithm is based on game theory, where all nodes participate in a simple game to choose an appropriate node willingness depending on its node density. We proposed that all nodes have two choices for its willingness: WILL_ALWAYS or WILL_NEVER.

We implemented our work in Riverbed Modeler 18.0 and compared it with the original OLSR protocol. Results show that our improved protocol outperformed the original OLSR in all performance metrics except end-to-end delay, which is negligible. The results from our simulations showed that our proposed algorithm could be feasible for real-time applications, especially multimedia applications in multihop networks.

However, for this paper, we did not consider the battery power of the nodes. It is known that OLSR consumes high power consumption, especially in finding alternative routes and keeping its route tables. Furthermore, another limitation is the choice of the mobility model. Random waypoint (RWP) was chosen as the mobility model. Using another mobility model and different parameters would be another perspective to observe our algorithm.

In our future work, it is necessary to extend this code to include more parameters like battery power. Most IoT devices are battery powered; thus power conservation is of utmost importance. It is also necessary to extend this code to consider Quality of Service (QoS) parameters, particularly for a high packet delivery ratio for multimedia application. Furthermore, we plan to implement our improved protocol on a testbed to test its efficiency.

**Author Contributions:** Conceptualization; O.O. and K.G.; Model design/experiment/analysis: O.O.; supervision: K.G. All authors have read and agreed to the published version of the manuscript.

**Funding:** This research received no external funding.

**Conflicts of Interest:** The authors declare no conflict of interest.

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
