# Peer review of "Improved Proactive Routing Protocol Considering Node Density Using Game Theory in Dense Networks"

_futureinternet, doi:10.3390/fi12030047_

Round 1

Reviewer 1 Report

There are numerous areas in which the paper needs to be improved. Some examples:

- The paper has several errors in grammar or spelling/vocabulary that distract the reader from the content.
- Research paper topics should be sufficiently informative and seamlessly integrated.
- The problem formulation and the description need to be improved to make the concepts clear and easy to understand.
- The paper structure should be improved. Section and subsection numbers need to be fixed.
- The introduction should end with an explanation of how the paper is structured.
- A related work section should be added. It should present a detailed discussion of the related work features and limitations to justify the presented proposal.
- Why Riverbed Modeler 18.0 was used for the simulation?
- The selection of values for input simulation parameters should be better justified.
- The simulation results were validated? Analytical results were validated against simulation results?
- 10 simulation runs are sufficient?
- Confidence intervals should be included, in addition to the average behaviour.
- More work is needed to assess the proof of concept of the proposed algorithm. How will the proposed algorithm perform in a different mobility scenario?
- Results should be compared with other research in this area.
- Some references are incomplete, abbreviated, or not properly formatted.
- Conclusions section should be rewritten to include main results and implications for future research.

Reviewer 2 Report

In this document, the authors apply game theory to reduce the control packets in dense networks using a proactive routing protocol.

The topic is within the scope of the Journal. 

In terms of the form, the paper is well organized. However, in the opinion of the reviewer, section 2 must be named "Materials and methods". Moreover, and most importantly, the paper must have a conclusion section.

The conclusion section must provide the most important findings for the reader, the limitations of the proposed method and the future work.

The related work must be a separated section. Numerous researchers have proposed several methods to improve routing protocol considering node density. Therefore, it is not clear about its contribution and value concerning the other works. Thus, the work should be improved, and the authors must clarify their contribution and value. 

The authors must add a table to compare the proposed approach with similar studies and state their contribution. This is the essential part to be improved on the document, and without this, the contribution of the paper is not clear.

Flowcharts must be added for the Algorithm 1-2. The limitations of the study are not presented. 

The reference section must be improved from the 33 references. I suggest these references from the "Future Internet" journal:

https://www.mdpi.com/1999-5903/10/2/16

https://www.mdpi.com/1999-5903/10/8/74

https://www.mdpi.com/1999-5903/11/1/18

https://www.mdpi.com/1999-5903/6/1/171

https://www.mdpi.com/1999-5903/10/5/41

I hope these suggestions will help improve your paper.

Round 2

Reviewer 1 Report

The authors have taken into account the reviewer's previous comments. As expected, the modifications improve the paper. Therefore, I propose that this paper should be accepted for publication.

Reviewer 2 Report

The reviewer is satisfied with the answers and revision provided by the authors.